# Unveiling the Intimate Mechanism of the Crocin Antioxidant Properties by Radiolytic Analysis and Molecular Simulations

**DOI:** 10.3390/antiox12061202

**Published:** 2023-06-01

**Authors:** Sarah Al Gharib, Pierre Archirel, Daniel Adjei, Jacqueline Belloni, Mehran Mostafavi

**Affiliations:** Institut de Chimie Physique, Université Paris-Saclay, UMR8000 CNRS, Rue Michel Magat, F-91405 Orsay, France; sarah.al-gharib@universite-paris-saclay.fr (S.A.G.); daniel.adjei@universite-paris-saclay.fr (D.A.); jacqueline.belloni@universite-paris-saclay.fr (J.B.)

**Keywords:** crocin, antioxidant properties, pulse radiolysis, gamma radiolysis, molecular simulation, mechanism

## Abstract

The successive steps of the oxidation mechanism of crocin, a major compound of saffron, by the free OH^•^ radical are investigated by pulse radiolysis, steady-state (gamma) radiolysis methods, and molecular simulations. The optical absorption properties of the transient species and their reaction rate constants are determined. The absorption spectrum of the oxidized radical of crocin resulting from the H-abstraction presents a maximum of 678 nm and a band of 441 nm, almost as intense as that of crocin. The spectrum of the covalent dimer of this radical contains an intense band at 441 nm and a weaker band at 330 nm. The final oxidized crocin, issued from radical disproportionation, absorbs weaker with a maximum of 330 nm. The molecular simulation results suggest that the OH^•^ radical is electrostatically attracted by the terminal sugar and is scavenged predominantly by the neighbor methyl site of the polyene chain as in a *sugar-driven* mechanism. Based on detailed experimental and theoretical investigations, the antioxidant properties of crocin are highlighted.

## 1. Introduction

Saffron is a spice derived from the flower of the plant *Crocus sativus*, commonly known as the “saffron crocus”, which can be classified as a potent antioxidant plant [1]. It is widely cultivated in countries around the world such as Iran, Greece, Italy, Spain, India, China, and Japan [2]. After collection and dehydration, it is used mainly for medicinal purposes [3,4], for food coloring, and as a flavoring agent. 

Saffron contains an impressive variety of plant compounds. The three main components of saffron are the crocin, the major one (30%), which accounts for the yellow pigmentation from the stigmas [5], the picrocrocin (5–15%), which gives the rusty, bittersweet flavor, and the safranal (up to 2.5%), the volatile oil, which lends the earthy fragrance to the spice [2]. The percentages of these compounds determine the quality and commercial grade of the saffron [6]. Moreover, saffron contains small amounts of vitamins, gums, proteins, amino acids, sugars, mineral matter, flavonoids, and other chemical compounds [7,8,9,10,11]. The chemical structure of crocin, that is, the diester formed from two saccharides gentiobioses and the dicarboxylic acid crocetin, is constituted of a conjugated chain of a polyene and of two terminal sugars, and it is presented in Figure 1.

The compounds of saffron are known to have strong antioxidant and radical scavenger properties against a variety of radical oxygen species (ROS) and pro-inflammatory cytokines. They give it medicinal properties and are used for a long time in traditional medicine for the treatment of different types of diseases [1].

Many studies mentioned that the health-promoting properties of saffron are attributed primarily to crocin, a unique carotenoid with a powerful antioxidant capacity [2]. Crocin scavenges free radicals, mainly OH^•^ radicals and superoxide anions, and so may defend cells against oxidative stress [1]. The crocin bleaching assay was also designed according to this important property of crocin as a basic element for the antioxidant activity of saffron [12,13]. The high molar absorptivity of crocin [14,15] allowed us to study crocin concentrations as low as 10 μM. 

The radiolytic systems ensure a selective and specific generation of the oxygen radicals OH^•^ (and/or O_2_^−•^ under aerated conditions), as well as of hydrated electrons e^−^_aq_. The crocin bleaching efficiency can, therefore, be correlated with the yield of these radicals. The pulse-radiolytic data demonstrated a very rapid diffusion-controlled attack by both hydroxyl radicals (OH^●^) or hydrated electrons (e^−^_aq_), whereas superoxide anions (O_2_^−•^) did not react at all [16]. Rate constants of OH^●^ and e^−^_aq_ with crocin were obtained using either a competition approach [17] (k_(OH_^●^_+croc)_ = 1.7 × 10^10^ M^−1^ s^−1^, k_(e−aq+croc)_ = 13 × 10^10^ M^−1^ s^−1^) [14] or a kinetic evaluation of the rapid bleaching by pulse radiolysis (k_(OH_^●^_+croc)_ = 3.33 × 10^10^ M^−1^ s^−1^ and k_(e−aq+croc)_ = 10.5 × 10^10^ M^−1^ s^−1^, respectively) [14]. 

One quantum chemical calculation only was published on the isolated crocin molecule [18], and several classical force field simulations about interactions of crocin with biological systems were performed [19,20]. No computational investigations of the absorption spectra of crocin, or of its radicals, have been published. However, several computational studies have been devoted to the structure and antioxidant properties of various other carotenoids [21,22,23,24,25,26], and to the conformation studies of fatty acids radicals [27].

Understanding the chemistry of mechanisms, advantages, and limitations of the methods is critical for the valid assessment of antioxidant activity in specific samples or conditions [28]. 

The aim of the present research is to study the reactions arising from the oxidation by OH^•^ radicals in the water of the crocin as the major compound of saffron by giving the detailed mechanism of OH^•^ attack and the nature of the products. The reaction rate constants of formation and decay of the radicals and their full absorption spectrum are measured by using the fast pulse radiolysis method and by following the transients. Based on the knowledge of the radiolytic OH^●^ yield in water, the spectra may be calibrated in molar absorption coefficients. In addition, as the crocin contains several sites for OH^●^ radical attack, simulations of the absorption bands of the radicals corresponding to these sites are performed in order to assign the structure of the radical formed from the comparison between the simulated and experimental absorption bands. The important role of the environment of the sites for the H-abstraction by OH^●^ radical is demonstrated. Finally, further products of crocin oxidation are also identified by giving the detailed mechanism of their formation. 

## 2. Materials and Methods

### 2.1. Solutions Preparation

Pure crocin was obtained from Sigma Aldrich Chemistry (Saint Quentin Fallavier, France). The nitrous oxide (N_2_O) was obtained from Air Liquide Industrial Gases Company (Limay, France). N_2_O is added to scavenge hydrated electrons and H^•^ radicals, enhancing the production of oxidizing OH^•^ radicals. Because of the well-known photosensitivity of the crocin, the solutions were freshly prepared in the dark before use, and the flasks were wrapped with aluminum films. The optical absorbance of samples was measured using a spectrophotometer Hewlett-Packard (Palo Alto, CA, USA). Given the difficulty of accessing very pure crocin, we checked the optical absorption properties of our samples. 

### 2.2. Radiolysis Experiments

Radiolysis (in the steady-state or the pulse regime) is a method delivering homogeneously known amounts of radicals. The oxidation study was performed using the panoramic ^60^Co source (2 kGy h^−1^) of ICP for steady-state irradiation, and the picosecond laser-induced electron accelerator ELYSE (Institut de Chimie Physique, Orsay, France) for pulse radiolysis [29,30]. The electron energy was 8 MeV, and the pulse length was 7 ps. The optical path of the cuvette was 0.5 cm, and the room temperature was 22 °C. The time-resolved optical absorption spectra were detected by using a Hamamatsu streak camera (Shizuoka, Japan). The UV-visible absorbances at 270–500 nm and the visible absorbances at 450–740 nm were detected by two different measurements. The uncertainty on the absorbance values was ±5%. The solution was refreshed between the pulses by using a circulating pump. The dose per pulse was derived from the absorbance of the hydrated electron in pure water with G(e_aq_^−^) = 3.3 × 10^−7^ mole J^−1^ at 5 ns and ε = 18,000 M^−1^ cm^−1^ at 660 nm [31,32]. 

The production of the oxidizing radicals OH^•^ arises from the radiolysis of the most abundant water molecules:H_2_O vvvv > e_aq_^−^ (2.8), H^+^ (2.8), H^•^ (0.62), H_2_ (0.47), OH^•^ (2.8), H_2_O_2_ (0.73)(1)
(in brackets are the yields in 10^−7^ mole J^−1^ units) [15]. 

These yields correspond to a scavenging factor of 10^−7^ s^−1^ (or 100 ns after the radical production and their mutual reactions in spurs). In N_2_O-saturated solutions, the crocin scavenges by Reaction (4) the OH^•^ radicals arising either primarily from the radiolysis of water (Reaction (1)) or from the scavenging by N_2_O of primary hydrated electrons and H^•^ radicals (Reactions (2) and (3)), respectively):N_2_O + e_aq_^−^ + H^+^  →  N_2_ + OH^•^(2)
N_2_O + H^•^  →   N_2_ + OH^•^(3)
croc + OH^•^  →  croc (–H)^•^ + H_2_O(4)

Reaction (3) plays a minor role because it is very slow and the yield of radical H^•^ is small. 

### 2.3. Molecular Simulations

UV-visible absorption spectra are often calculated through geometry optimization and harmonic frequency calculations of both the ground and excited states. This is the method developed by Brémond et al. who were able to address in this way large organic molecules, with an explicit introduction of vibronic couplings [33,34].

Over the last few years, we have developed a different method based on molecular simulations of the species of interest: (i) the potential surface of the solute is investigated with the help of the classical MC (Monte Carlo) method [35]; (ii) at each MC step the electronic structure of the solute is calculated with the DFT (density functional theory) method and the lc-ωPBE functional [36]; (iii) the solvent is modeled with the help of the PCM (polarized continuum model) method, in its SMD (Solvent Model using Density) variant [37], (iv) the absorption spectrum is given by the convolution of a list of TDDFT (time dependent DFT) lines calculated on a sublist of MC configurations using the formula from the literature [38]. This sublist is built with uniform sampling, selecting every 100th step of the MC simulation. This method enabled us to obtain useful insights into the mechanisms of radiolytic processes in solution [39,40,41].

In the present simulations, we use two MC moves: (i) multi-stretch moves, operating collective displacements of all the atoms [32], with probability of 0.8, and (ii) torsional moves about all the chemical bonds with probability of 0.2. For critical species, we have performed simulations with 60,000 MC steps and discarded the first 20,000 steps for the calculation of the spectrum. For other species, we simulated with 20,000 MC steps only. The convolution of TDDFT lines uses a Gaussian function with a given *fwhm* parameter (full width at half maximum), to be discussed.

We also considered spin contamination in DFT results [36]. This phenomenon is due to the one-determinant DFT wave function [42]. In the present work, we generalize the *spin screening* procedure we already used for moderating this drawback of DFT [36]: in the list of TDDFT lines contributing to the absorption spectrum, all the lines with a <S^2^> value larger than the value in the ground state by some amount are discarded. This amount, the so-called spin screening parameter, will be discussed.

### 2.4. Thermochemistry

The thermochemical quantities, such as association free energies, barrier heights, and oxidation processes, have been performed with the Gaussian package [34], with the lc-ωPBE functional [33], and also ordinary DFT functionals such as B3LYP, and cam-B3LYP, and with the *SMD* method [34]. For the simulations, we used the small *SDD* (Stuttgart Dresden) Gaussian basis set. This small *SDD* basis has been provided with *d* polarization functions on C atoms (exponent 0.83) and O atoms (exp. 1.), and the resulting basis set will be called *pSDD* (polarized *SDD*) hereafter. For a few calculations, we also provided the *pSDD* basis set with diffuse functions on C atoms: *s* (0.07), *p* (0.08) and *d* (0.55, 0.15), and on H atoms: *s* (0.3, 0.1), and the resulting basis will be called *pSDD+* hereafter. Thermochemical calculations were done with the large *aug-cc-pvdz* basis set [34]. This large basis will be called *apvdz* hereafter.

Association free energies are difficult quantities, and we propose approximate values with the help of the Wertz formula [43]: Δ_r_G^pcm^_corr_ = Δ_r_G^pcm^ − 0.5 (Δ_r_G^gp^ − Δ_r_H^gp^) + RT(5)
where Δ_r_G^pcm^ is the crude PCM value of the association free energy and Δ_r_G^gp^ and Δ_r_H^gp^ are the gas phase values. The +RT term cancels the PV contribution to enthalpy in the Gaussian code. This method is efficient for covalent dimers [44].

For van der Waals dimers, we also included in the Δ_r_G^pcm^ term of Formula (5) dispersion interactions, and correction of the basis set superposition error (BSSE) [34]. Structures were visualized and processed with the GaussView code [45].

Crocin displays two bulky sugar substituents (Figure 1), making DFT simulations unworkable. Alternative solutions could be the use of QMMM or DFTB methods [34], which would accelerate the quantum calculations, but make the simulations much too long. We, therefore, modeled crocin by the di-methanol ester of crocetin (Me_2_–crocetin), by replacing the two gentiobiose groups of crocin with two methyl groups (Figure 1). The specific behavior of these crocin sugar groups was considered separately. This approach will be firmly justified later.

We finally optimized the ω parameter of the lc-ωPBE functional, so as to reproduce for Me_2_–crocetin at 0 K the measured value of λ_max_ = 441 nm for crocin. Optimization yields the value *ω* = 0.244 and λ_max_ = 441 nm, *f* = 3.22.

## 3. Radiolytic Oxidation Results and Discussion

### 3.1. Solubility of the Crocin in Water

We observed the optical absorption spectra of the samples at various concentrations to be compared with the literature data. The absorptivity of crocin is very intense, and the optical path of the cell was 1 or 2 mm. The crocin spectrum is constituted of an intense band with a maximum of 441 nm, a shoulder at 460 nm, and two weak UV bands at 250 and 333 nm (Appendix A) [46]. No absorbance is detected beyond 550 nm. Two studies evaluated the molar absorption coefficient of crocin at the maximum of 441 nm: ε_441_(croc) = 1.369 × 10^5^ M^−1^ cm^−1^ [14], and ε_441_(croc) = 1.335 × 10^5^ M^−1^ cm^−1^ [13]. The average value of ε_441_(croc) = 1.35 × 10^5^ M^−1^ cm^−1^ will be used in the present work.

The absorbance increases proportionally to the concentration up to [croc] = 0.08 mM. But, noteworthy, beyond 0.13 mM, the absorbance is almost constant (Appendix A, inset). The crocin solubility in water is, therefore, limited to this value. 

### 3.2. Reaction of Crocin with OH^•^ Radical within 1 μs

Kinetics of the reaction. Figure 2a presents the pulse radiolysis results within 1 μs of the difference spectrum arising from the OH^•^ scavenging by the crocin at the concentration of 0.0385 mM. At 1 μs, all OH^•^ radicals formed in Reactions (1) and (2) have reacted (Reaction (3) is too slow to contribute). A bleaching band is increasing with time at 441 nm at the same wavelength as that of the crocin and with the same width, as already observed [16]. In addition, a new absorption band is observed with a maximum of 678 nm, which increases with time up to 1 μs. Noteworthy, two isosbestic points are observed at 375 and 510 nm. They indicate first that the reaction product arises directly and stoichiometrically from the crocin oxidation by the OH^•^ radical and that it must be assigned to the oxidized radical R^•^ = (croc (–H))^•^. Secondly, this radical absorbs at these isosbestic points with the same absorptivity as the crocin, but less in the interval 375–510 nm where bleaching is observed (Figure 2). Around 333 nm, where the crocin spectrum contains a weak band, the absorbance is slightly enhanced in the differential radical spectrum. These optical features of the radical at 678, 441, and 333 nm were not yet mentioned in the literature. That at 678 nm will be crucial to conclude from molecular simulations (see Section 4.1) on the site of the H-abstraction by OH^•^ radicals.

Figure 2b presents the kinetics in the range of 1 μs of the negative differential absorbance (bleaching) at 441 nm where it increases, of the positive absorbances at 678 nm and 333 nm where it increases, and at the isosbestic wavelengths 375 and 510 nm where it remains zero. In addition, a bump of a rapid increase soon after the pulse, followed by a decrease at 50 ns, is observed on each curve. It is assigned to the formation, within the pulse, of the hydrated electrons which then react with N_2_O and yield OH^•^ radicals in <10 μs.

The transient differential spectra within 500–750 nm at 1 μs and the kinetics at 678 nm are compared in Appendix A for various crocin concentrations between 0.023 and 0.13 mM. Similar features are found for these concentrations. 

The pseudo-first-order test applied to the absorbance A_678_ increase at 0.13 mM is a straight line (Appendix A). The second order rate constant of the Reaction (4) of OH^•^ with crocin is thus k_croc+OH•_ = 3.4 × 10^10^ M^−1^ s^−1^. This value is close to the literature one (k_croc+OH•_ = 3.33 × 10^10^ M^−1^ s^−1^) [16] and confirms that the crocin is an efficient antioxidant to scavenge the radicals OH^•^. In fact, the particularly high k_croc+OH•_ value between neutral species favors crocin in the competition with other molecules, even more concentrated, if their scavenging rate constant is much lower than k_croc+OH•_. The crocin efficiency as an antioxidant is still more significant if it would prevent the deleterious effect of induction by OH^•^ of oxidation chain reactions of neighbor molecules with oxygen.

### 3.3. Calibration in Molar Absorption Coefficient of the Radical Optical Spectrum

At the dose of 97 Gy per pulse with a yield of G(OH^•^) = *G_eaq−_* + *G_OH_•* = 5.6 × 10^−7^ mol J^−1^, the concentration of OH^•^ radicals is [OH^•^] = 5.4 × 10^−5^ M per pulse. We conclude that after reaction with OH^•^ radicals at 0.06 mM (Reaction (4)), the crocin at the initial concentration of 0.0385 mM is totally oxidized into the radical (croc (_–_H))^•^. The differential spectrum at 1 μs in Figure 2 is thus the difference between the spectrum of the radical and that of the initial crocin, whose initial absorbance for 1 mm optical path was A_441nm_(croc) = 0.52. From the stoichiometry indicated by the isosbestic points and the ratio of the respective absorbances of crocin at 441 nm and radical at 678 nm with the equal optical path, we derive the molar absorption coefficient at 678 nm of the radical: ε_678nm_(croc (–H))^•^ = ε_441nm_(croc) × A_678nm_(croc (–H))^•^/A_441nm_(croc) = 6230 M^−1^ cm^−1^. (Table 1 and Figure 3b).

At 441 nm, the transient differential absorption spectra of the radical allows bleaching relative to the initial intense absorption band of the crocin (Figure 2a). The negative absorbance of the bleaching at 441 nm and 1 μs is almost three times higher than the positive absorbance at 678 nm (Figure 2). However, it is clearly much smaller than the expected bleaching arising from the consumed crocin corresponding to the OH^•^ scavenging. Therefore, the radical also absorbs at around 441 nm, but less than crocin. It appears that, in addition to the supplementary band around 678 nm, the radical spectrum is very similar to that of the crocin, though less intense in the interval between the isosbestic wavelengths 375 and 510 nm (Figure 3a). A previous study had also reported the increase in bleaching at 440 nm, much less intense than expected from the OH^•^ scavenging, and had concluded just a partial OH^•^ scavenging, of a few percent only [15]. The simultaneous increasing absorbance arising from the radical, also at the same wavelength range as the crocin, had not been considered, and the conclusion deduced about the OH^•^ scavenging efficiency was wrong. At 441 nm, the differential molar coefficient Δε_441nm_ = ε_441nm_ (croc) − ε_441nm_(croc (–H))^•^ = 18,200 M^−1^ cm^−1^, that is, ε_441nm_(croc (–H))^•^ = 1.17 × 10^5^ M^−1^ cm^−1^. At 333 nm, Δε_333nm_ = ε_333nm_(croc (–H))^•^ − ε_333nm_ (croc) = 3000 M^−1^ cm^−1^, that is, ε_333nm_(croc (–H))^•^ = 0.10 × 10^5^ M^−1^ cm^−1^ (Table 1). The complete radical spectrum is obtained from the algebraic sum of the crocin spectrum and of the differential spectrum of the radical, both calibrated in molar absorption coefficients (Figure 3).

### 3.4. Reaction of the Crocin Oxidized Radical (R^•^) within 1 to 500 μs

*Reaction kinetics.* At ≥1 μs, all the OH^•^ radicals have been consumed (by scavenging or dimerization), and the formation of the radical is over. Then, beyond 1 μs, it is observed that the absorbance at 678 nm of the crocin oxidized radical decreases with time. Figure 4 presents the spectra evolution beyond 1 μs for [croc] = 0.10 mM and the kinetics of this decay over 84 μs. For better accuracy, the curve summarizes the absorbance data at 0.8 μs (Figure 4b), and of the decay kinetics over 10 and 84 μs (insets) (without the initial rapid increase in the radical formation).

The absorbance at 678 nm completely vanishes at 500 μs (Appendix A), meaning that the product arising from the decay reaction of two radicals does not absorb at this wavelength and that the absorbance must be assigned exclusively to the (croc (–H))^•^ radical.

The second-order test of the radical decay at 678 nm is given in Figure 4b, inset. The test in 1/At is a straight line, indicating that two radicals react with a rate constant k_obs_ = 0.6 × 10^6^ s^−1^. According to the molar absorption coefficient derived above ε_678_ (croc (–H))^•^) = 6.23 × 10^3^ M^−1^ cm^−1^, the second order rate constant of the radical decay is 2k_R+R_ = = 3.7 × 10^9^ M^−1^ s^−1^. This reaction could be assigned to a dimerization or to a disproportionation of the radical. However, this diffusion-controlled process during 80 μs corresponds better to the radical dimerization (see the Simulation section):2(croc (–H))^•^  →  (croc (–H))_2_(6)

### 3.5. Calibration of the Optical Absorption Spectrum of the Radical Dimer

During the first μs, the absorbance at 678 nm and the bleaching at 441 nm both increase with a constant ratio because they belong to the same radical spectrum (Figure 2b). In contrast, within 1 and 420 μs, the absorbance at 678 nm decays to zero, whereas the 441 nm bleaching still increases, but more slowly, up to a plateau where ΔA_441_(420 μs) = −0.74 (Figure 5b). The kinetics at 441 nm is a convolution of decay of the radical absorbance, similar to that at 678 nm, and of an increase in a more intense radical dimer bleaching, arising by Reaction (5). Another small increase is observed in the positive 330 nm band. Note that, during the supplementary increase, the shape of the bleaching band changes with a maximum wavelength shifting from 441 nm to 460 nm. No isosbestic point is observed.

The concentration of (croc (–H))_2_ is half of that of the radical precursor, and the molar absorption coefficient of the dimer is thus double that of one monomer. Therefore, the molar absorption coefficient at 441 nm of the radical dimer ε_441_(croc (–H))_2_ is deduced from the ratio between the differential absorbances at 1 and 420 μs by the following equation: [ε_441_(croc) − ½ ε_441_(croc (–H))_2_]/[ε_441_(croc) − ε_441_(croc (–H))^•^] = ΔA_441_(420 μs)/ΔA_441_(1 μs)
that is, ε_441_(croc (–H))_2_ = 1.84 × 10^5^ M^−1^ cm^−1^ (Table 1). The complete dimer spectrum is obtained from the algebraic sum of the crocin spectrum and of the differential spectrum due to the dimer with the bleaching band at 420 μs, both calibrated in molar absorption coefficients (Figure 5b). The spectrum of the radical dimer is more intense (though half of it is less intense) than those of the crocin, or of the radical, and the shoulder at 460 nm is less noticeable. At 333 nm, ε_333_(croc (–H))_2_ = 0.31 × 10^5^ M^−1^ cm^−1^ (Table 1).

### 3.6. Gamma Irradiation of Crocin

A crocin solution at 0.057 mM was gamma-irradiated at an increasing dose of up to 211 Gy (Figure 6). The main optical absorption band at 441 nm and the small band at 230 nm of the crocin decrease in intensity while keeping the same shape. Simultaneously, a new band with a maximum of 330 nm increases. An isosbestic point is seen at 375 nm up to the dose of 97 Gy. Beyond 100 Gy, the isosbestic point disappears. The dose-dependence of the absorbance at these three wavelengths is presented in Figure 6a, inset.

Note that, under the conditions of the γ-irradiation, the end spectrum is observed in the range of several minutes, though the dose is of the same magnitude as in one electron pulse. Another main difference with pulse radiolysis is that the OH^•^ radicals are produced and scavenged steadily and that their concentration is much lower than in pulse radiolysis so that the OH^•^ scavenging is not in competition with their dimerization. Therefore, OH^•^ radicals are completely scavenged by crocin with the oxidation of crocin into the radical (croc (–H))^•^, dimerizing into (croc (–H))_2_ (Reaction (6)) as a transient, then into the molecule (croc (–H_2_)) with 2 H-atoms abstracted, by a disproportionation reaction:(croc (–H))_2_  →  (croc (–H_2_)) + croc(7)

The radical (croc (–H))^•^ resulting from the OH^•^ scavenging is also in low concentration. As well, it might first associate with the crocin into the complex (croc, croc (–H))^•^ (Reaction (8)), before yielding the same oxidized molecule as above by direct disproportionation of this complex (Reaction (9)):(croc (–H))^•^ + croc  →  (croc, croc (–H))^•^(8)
2(croc, croc (–H))^•^  →  3 croc + (croc (–2H))(9)

The dose-dependent spectra are constituted of two components, the decreasing crocin (with the molar absorption coefficient ε_441_(croc) = 1.35 × 10^5^ M^−1^ cm^−1^) and, after the disproportionation (9), the increasing reaction product assigned to the molecule (croc (–2H)). From the deconvolution of the dose-dependent spectra in γ-radiolysis up to 97 Gy (Figure 6b), the maximum of the (croc (–2H)) spectrum is at 330 nm. The molar absorption coefficients are ε_330_(croc (–2H)) = 0.25 × 10^5^ M^−1^ cm^−1^ and ε_441_(croc (–2H)) = 0.11 × 10^5^ M^−1^ cm^−1^. The isosbestic wavelength with the crocin is ε_375_(croc (–2H)) = 0.18 × 10^5^ M^−1^ cm^−1^ (Table 1). The intensity of this spectrum is weak compared to those of crocin or of the radical dimer. This point will be discussed in Section 5 below. Beyond 100 Gy, crocin is partly exhausted, and the accumulated product (croc (–2H), also containing methyl sites, is now attacked by OH^•^, therefore it plays the same role as an antioxidant. By this oxidation, a secondary product is formed with molar absorption coefficients at 330 and 375 nm lower than for croc (–2H) (Figure 6a, inset).

From the absorbance decrease at 97 Gy (ΔA_441_ = 0.49) and the differential molar absorption coefficient (Δε_441_ = ε_441_ (croc) − ε_441_ (croc (–2H)) = 1.24 × 10^5^ M^−1^ cm^−1^), the experimental consumption yield of crocin in N_2_O-saturated solutions irradiated by γ-rays (Figure 6, inset) is G_exp_(- croc)(γ) = 4.0 × 10^−7^ mole J^−1^ (Table 1). The hydrogen peroxide produced by the water radiolysis (Reaction (2)) is also able to oxidize slowly crocin during the γ-irradiation [14]. The calculated overall oxidation yield after the disproportionation would be equal to G_calc_(- croc)(γ) = G_calc_(croc (–2H))(γ) = ½ (Ge_aq_^−^ + GOH^•^ + GH^•^ + 2GH_2_O_2_) = 3.84 × 10^−7^ mole J^−1^ (Scheme of Figure 7b and Table 1). This value is slightly lower than the experimental scheme above, possibly because of some simultaneous photodegradation of crocin. The agreement between G_calc_(- croc)(γ) and G_exp_(- croc)(γ) and the existence of isosbestic points during the oxidation support the occurrence of the disproportionation Reaction (8). Instead, in pulse radiolysis, the dimerization is predominant and the oxidation by H_2_O_2_ is too slow and does not yet occur: G_calc_(croc(–2H))(pulse) = ^1^/_2_G(- croc) = 3.1 × 10^−7^ mole J^−1^ (Scheme of Figure 7 and Table 1).

The silver ion reduction is an adequate way to check the possible antioxidant properties of the radical (crocin (–H))^•^. For that purpose, a mixed solution of 0.1 mM of crocin and 0.1 mM of silver perchlorate was irradiated by γ-rays. The silver cations scavenge the H^•^ radicals faster than N_2_O and are reduced into silver atoms (Reaction (9)).
Ag^+^ + H^•^ →  Ag^0^ + H^+^(10)

It is observed (Figure 8a) that the absorption spectrum decreases at increasing doses with an isosbestic wavelength at 375 nm without Ag^+^. However, the bleaching yield of the crocin at 441 nm is much less in the mixed silver and crocin solution than in the solution without silver (Figure 6). Therefore, the radical (crocin (–H))^•^ does not reduce Ag^+^ into a silver atom. The reduction potential of this radical E^0^(croc (–H_2_))/(croc (–H))^•^ is thus much less negative than that of the radical of formate CO_2_^−•^ [2], or of citrate (cit –(H))^•^ [2], which did reduce the silver ions (E^0^(Ag_2_^+^/2Ag^+^) = −1.2 V_NHE_) and enhanced the antioxidant properties of their precursors.

Moreover, to explain that the observed yield with Ag^+^ G_exp_(- croc)_Ag+_ = 2.1 × 10^−7^ mole J^−1^ is even less than the value without Ag^+^ (G_exp_(- croc) = 4.0 × 10^−7^ mole J^−1^) (Table 1), we suggest that the silver atoms arising from the H^•^ scavenging do reduce the crocin into the radical (croc^−•^), which itself recombines with the oxidized radical (croc –(H))^•^ into two crocin molecules (Scheme of Figure 8b):Ag^0^ + croc    →    Ag^+^ + croc^−•^(11)
croc^−•^ + croc (–H) + H_2_O    →  2 croc + OH^−^(12)

The silver ions also catalyze the hydrogen peroxide decomposition. After complexation and disproportionation of the radicals (Reactions (8) and (9)), the calculated value G_calc_(- croc)_Ag+_ = G_eaq_^−^ + G_OH•_ − G_H•_ = 2.5 × 10^−7^ mole J^−1^ is in agreement with the experimental value (Table 1).

## 4. Thermochemistry of Crocin and Me_2_–Crocetin

### 4.1. Oxidation Free Energies

We considered all the possible radicals yielded by H abstraction from the model Me_2_–crocetin according to:Me_2_–crocetin + OH^•^  →   (Me_2_–crocetin (–H))^•^ + H_2_O(13)

The reaction free energies have been calculated at the lc-ωPBE (0.244)/*apvdz*/*SMD* level. Results for the H-abstraction from C_a_ and C_b_ carbons are given in Table 2 (the results for the symmetrical C_a_′ and C_b_′ carbons are obviously the same as for C_a_ and C_b_). They have been obtained with the most stable conformer of each species. Results for abstraction from other species: –CH groups of the polyene chain and for the cation formation are given in Appendix A. It can be seen in Table 2 that oxidation is very efficient for the C_a_ and C_b_ carbon atoms, with reaction free energies close to −2 eV, and C_a_ is slightly more oxidable than C_b_. Other carbon atoms, belonging to the polyene chain, are clearly less oxidable, with reaction free energies ranging between −1.6 eV (C_3_) and −1.2 eV (C_4_) (Appendix A).

In the following we note R^•^C_a_ and R^•^C_b_ the (Me_2_–crocetin (–H))^•^ radicals after H-abstraction from the –C_a_H_3_ and –C_b_H_3_ groups.

Note that each crocin molecule contains four methyl groups and that their high affinity for OH^•^ radicals could explain the high value of the reaction rate constant per molecule measured above by pulse radiolysis. Moreover, according to the free energy (−2 eV) of the Me_2_–crocetin oxidation reaction to R^•^C_a_ or R^•^C_b_ (Table 2), and to the redox potential of OH^•^ radicals (E°(OH^•^/H_2_O) ≈ 2.0 V_NHE_ at pH 7), the redox potential of Me_2_–crocetin is E°(Me_2_–crocetin (–H)^•^/Me_2_–crocetin) ≈ 0 V_NHE_. This means that as far as crocin may behave as Me_2_–crocetin, it is able also to play an indirect role of antioxidants in restoring, by H-atom transfer, oxidized neighbor molecules with positive redox potentials.

### 4.2. Transition States

Transition states (TS) have been calculated for the oxidation of *β*-carotene by the NO_2_ radical [20]. *β*-carotene has the same polyene chain as crocin, with four methyl groups, but very different end-of-chain groups. The authors found the following barriers to *β*-carotene oxidation: +0.45 eV for –C_a_H_3_ and +0.50 eV for –C_b_H_3_ (using the numbering of Figure 1). They also found lower barriers for the oxidation of the end groups. Calculations on a series of carotenoids show that oxidation barriers are always smaller on the end groups [22]. 

We found the calculation of some TS of Me_2–_-crocetin very delicate due to very small vibration frequencies. Nevertheless, we could obtain reliable values through the comparison of Me_2_–crocetin with two smaller systems: model 1, which has a much shorter polyene chain (Figure 9), and model 2, which has no COO groups (Appendix A). We also compared the small *pSDD* and large *apvdz* bases. The discussion is given in Section S4.2, results are shown in Appendix A, and summarized in Figure 9. 

TS free energies for the –CH groups are easily calculated. Figure 9 shows that the lower barriers for C_3_, C_5_, and C_7_ amount to ≈0.1 eV in model 1. In Appendix A we show the basis effect in model 1 and deduce that in Me_2_–crocetin these barriers are a little bit smaller, but remain positive. TS free energies for the –CH_3_ groups proved very delicate. In model 1, we had to force harmonic analysis with residual nuclear gradients ≈ 10^−4^ a.u., larger than the usual threshold and obtained negative values ≈ −0.4 eV. We could confirm this negative sign with the help of model 2 where both –COOCH_3_ groups have been replaced by two –CH_3_ groups (Appendix A). In model 2, the negative value −0.6 eV for –C_a_H_3_ was easily and securely obtained. Negative TS energies are not surprising: they mean that the free energy curve is attractive at longer distances and that the TS lies under the asymptote. Note that this negative value is due to the OH^•^ radical: Appendix A shows that the NO_2_ radical gives the positive free energy: +0.3 eV to this same TS in model 2, in agreement with the value of Ref. [20].

Structures of a few TS are shown in Appendix A. They explain the issues encountered by the –CH_3_ groups. The ”easy” TS of –CH groups is of a stretching type, with large imaginary frequencies, whereas the TS of the –CH_3_ groups are of a bending type, with OH pointing toward one O atom, provoking very small imaginary frequencies. 

We conclude that the H abstraction from the methyl groups –C_a_H_3_ and –C_b_H_3_ is easily done, without barrier, and that this abstraction from the –CH groups of the polyene chain is scarcely possible at 22.5 °C. This is consistent with the reaction free energies of Table 2 and Appendix A. Besides, the H abstraction by the OH^•^ radical from a glucose molecule with the same method was also investigated. All these calculations yielded high potential barriers: +0.29 eV and +0.33 eV for the carbon and oxygen atoms of the –CH_2_OH group, and + 0.36 eV for –OH groups of the cycle. These numbers confirm that the sugar groups of crocin are much less oxidable than the methyl groups of the chain.

### 4.3. Conformation Analysis

It is well known from the literature that *cis–trans* isomerization can be critical for molecules with a polyene chain, such as carotenoids [24] or fatty acids [25]. In the following, we call *all-trans* the structure of Figure 1, with no *cis–trans* isomerization and (x,y) the *cis–trans* isomerization about the C_x_C_y_ bond. We first investigated the *cis–trans* issue for the crocin and Me_2_–crocetin molecules, optimizing the seven structures corresponding to one *cis–trans* isomerization about one bond. In Appendix A, we give the free energies and low energy optical absorption data for the *all-trans* and the first three *cis–trans* conformations lying under ≈0.1 eV. It can be seen that crocin and Me_2_–crocetin behave similarly as long as conformation and optical absorption are concerned. 

Both molecules are mainly *all-trans*, with the lowest (6,7) *cis–trans* isomer 0.07 eV higher. Both other (2,3) and (8,8′) conformations also interfere. In Me_2_–crocetin a very constant 5–6 nm redshift, with respect to crocin, is observed in every conformation. This shift is very small. In both molecules, the optical data hardly depend on the conformation, with again a 5–6 nm shift between conformations.

For both radicals R^•^C_a_ and R^•^C_b_, we used a more demanding method: (i) optimization of the 13 conformers involving one *cis–trans* isomerization about the 13 chemical bonds of each radical, (ii) 13 MC simulations at 1000 K, initialized with each of these optimized structures, and (iii) geometry optimizations and harmonic analysis from a sub-list of 130 MC conformations of these simulations. This procedure enables the test of multiple *cis–trans* isomerizations. The free energies and first excited states are given in Appendix A.

The R^•^C_a_ radical also has a dominant *all-trans* conformation, but the bent conformers (5,6), (2,3), and (2′,3′) lie only 0.03, 0.5, and 0.08 eV higher. One intense transition at 460–470 nm can be seen, and two weak, lower energy transitions interfere. The R^•^C_b_ radical displays two quasi-degenerate conformers, the *all-trans* and the (6,7) bent ones. Two other bent conformers, (5,6) and (5,6) (6,7), lie only 0.025 and 0.05 eV higher. Therefore, a double *cis–trans* isomerization interferes. One intense transition at 410–420 nm can be seen, and three weak, lower energy transitions interfere.

The absorption behavior of the radicals is a critical issue. In Appendix A, the absorption lines of the optimized structures of Appendix A have been broadened in order to mimic absorption spectra. Appendix A shows that *cis–trans* isomerization modifies the absorption spectra very slightly.

## 5. Molecular Simulations

The simulated optical absorption spectra may be directly compared with the observable time-resolved or final spectra of radiolysis. When fitting with the experimental ones, they give essential insights into the mechanisms of oxidation. It was mentioned in Section 2.3 that the extraction of absorption spectra from molecular simulations is conditioned by the *fwhm* parameter, that is, the broadness of the Gaussian function used for the convolution of TDDFT lines, and by the spin screening parameter, used for the empirical reduction of spin contamination in radical species. The choice of these parameters is discussed in Section S5 and Appendix A in the case of the R^•^C_a_ radical. According to this discussion, three values of the *fwhm* parameter will be used: *fwhm* = 0.7 eV for the broadening of TDDFT lines of optimized structures, *fwhm* = 0.50 eV for the broadening of TDDFT lines of the intense bands, and for the weak, high-energy bands of the simulation, and *fwhm* = 0.20 eV for the weak, low energy bands of the simulation. The spin screening parameter will be 20%.

### 5.1. Absorption Spectrum of Me_2_–Crocetin

The absorption spectrum of crocin, obtained experimentally (Appendix A) is rather complex, with two weak, high-energy bands at 260 and 320 nm and an intense, dissymmetrical, and structured band at 441 nm. The simulated spectrum of Me_2_–crocetin, unconstrained and initialized with the *all-trans* conformer, is shown in Figure 10a, calculated with the pSDD basis, the fwhm value 0.50 eV, and 60,000 MC steps. This spectrum displays some of the features observed for crocin: the weak bands at 260 and 320 nm, and an intense band at 448 nm. However, this intense band is symmetrical and displays no structure. Figure 10b shows the same spectrum, calculated with the larger pSDD+ basis, but this only produces a slight redshift. This raises the question of whether the discrepancies between measurement and simulation are due to a defective simulation of Me_2_–crocetin, or to our modeling of crocin by Me_2_–crocetin.

The answer to this question is given by the conformation analysis: in fact, the isomerization free energies of Appendix A: 0.07, 0.08, and 0.11 eV, are rather large, but the DFT calculations and harmonic analysis make them rather inaccurate. Actually, these free energies only suggest that the interference of bent conformers cannot be excluded. Therefore, we have performed *constrained* 20,000 steps MC simulations of all the bent conformers of Me_2_–crocetin, with only atom multi-stretch moves and no dihedral angle fluctuation. Typical structures of the most stable four conformers of Me_2_–crocetin according to Appendix A, namely the *all-trans* conformer and the (2,3), (6,7), and (8,8′) bent conformers, are shown in Appendix A. Figure 11a presents their spectra. It can be seen that (i) the weak band at 250 nm can be attributed to the *all-trans* and (2,3) bent conformers, (ii) the weak band at 320 nm to the bent (6,7) and (8,8′) conformers, (iii) the *all-trans*, (2,3) bent and (6,7) bent conformers have very close maxima: 438, 444, and 440 nm, respectively, but the bent (8,8′) conformer at 451 nm undergoes a 13 nm red shift compared to the *all-trans* conformer. The conformers’ spectral features of Me_2_–crocetin (Figure 11a) provide a plausible interpretation of the shape of the experimental spectrum of crocin in Figure 11b. In particular, the striking shoulder on the red wing of the spectrum is assigned to the (8,8′) conformation.

Note that the lengthy simulated, *unconstraint* spectrum of Figure 10a, with λ_max_ = 448 nm is close to the constraint, *all-trans* spectrum of Figure 11a with λ_max_ = 438 nm. This unconstraint simulation has been initialized with the *all-trans* conformer and keeps it all the time. According to the large free energies of Appendix A, this is quite normal. Therefore, the present unconstrained simulation, even if extended, cannot yield the structure of the observed spectrum. The right Monte Carlo treatment of these conformation issues would be the *parallel tempering* Monte Carlo method [32], with several parallel, coupled MC simulations at different temperatures. However, the analysis of Section 4.3 shows that parallel tempering is not necessary and that ordinary, unconstrained MC simulations of the main conformers provide a realistic investigation of the absorption spectra. 

We conclude that Me_2_–crocetin is an adequate model of crocin. The present results for Me_2_–crocetin, including the interference of *cis–trans* conformers, emphasize the predominant influence of the polyene chain on the optical spectrum. Significantly, they may explain not only the spectrum of crocin but also the spectrum shape of *β*-carotene and of numerous carotenoids, mainly the same intense and dissymmetrical band [47]. 

### 5.2. Absorption Spectra of Me_2_–Crocetin Radicals

The spectra of all the possible oxidation products of Me_2_–crocetin were then simulated. Since the reaction barriers of Figure 9 show that the methyl groups are easily oxidable, the spectra of the R^•^C_a_ and R^•^C_b_ species are given in more detail. In Figure 10 the focus is made on the intense bands with *fwhm* = 0.50 eV and the small *pSDD* (a) and large *pSDD+* (b) bases, and in Figure 12 we focus on the weak bands only, at higher (a) and lower (b) energies with the large basis only. We label “½ ½” the half-sum of the spectra of R^•^C_a_ and R^•^C_b_ species, assuming equal oxidation probabilities, and “^5^/_6_ ^1^/_6_” another combination of the spectra assuming that R^•^C_a_ is predominant, these numbers will be explained. The spectra of the R^•^C_2_, R^•^C_3_, R^•^C_4_, R^•^C_6_, and R^•^C_7_ species are given in Appendix A, together with that of the cation. 

We now compare Figure 3a,b of the experimental spectrum and Figure 10 and Figure 12 of the simulated spectrum, respectively. Figure 3a shows that the oxidation product has an intense absorption band very similar to that of crocin, with a slight decrease in intensity. Figure 10a shows that the two possible products, R^•^C_a_ and R^•^C_b_, have an intense band with very different maxima at 467 nm and 415 nm, respectively. From the large blue shift of 33 nm (0.20 eV) and the large absorption intensity decrease obtained for R^•^C_b_ alone, it appears that this radical is not predominant in contrast with R^•^C_a_, yielding a slight decrease in intensity and a 20 nm redshift (0.10 eV). If R^•^C_a_ and R^•^C_b_ were equally formed (“½ ½” hypothesis), they would yield a larger decrease in intensity and a tiny 5 nm blue shift. Using the larger basis does not change this situation (Figure 10b).

Figure 12a (to be compared with Figure 3b) shows that the weak, high-energy band observed at 330 nm must be attributed to R^•^C_a_ only. The simulated band (in red) lies at 343 nm (error: 13 nm, 0.1 eV). Figure 12b shows that the observed weak, low-energy absorption band must be also attributed to R^•^C_a_ only. Note that the observed maximum lies at 678 nm and that the simulated values are 646 nm with the small basis (not shown, error: 0.1 eV) and 654 nm with the larger basis (error: 0.05 eV). Figure 12b also shows that the little bump at 620 nm, which is present in the measured spectrum, may be attributed to R^•^C_b_ alone. The simulated band lies at 592 nm (error: 0.10 eV). This bump appears to be the only direct interference of R^•^C_b_ in the spectrum. In brief, Figure 10 and Figure 12 suggest that R^•^C_a_ is predominant, and R^•^C_b_ observable but rare, in contradiction with thermochemistry which rather implies equal probabilities of the two radicals and the ½ ½ spectra of Figure 10 and Figure 12. The predominant formation of R^•^C_a_ now requests supplementary arguments.

We first note that the real crocin molecule offers to the OH^•^ radical a polyene chain and two bulky sugar substituents of comparable sizes. The probability that OH^•^ first meets the polyene chain amounts to ≈^1^/_3_ only. Therefore, and according to thermochemistry, the oxidation probabilities of R^•^C_a_ and R^•^C_b_ are the same: ½ × ^1^/_3_ = ^1^/_6_. 

We then consider that, arriving on a sugar substituent, the OH^•^ radical might undergo an electrostatic attraction by the sugar. Actually, the partial charges of the oxygen atoms of the gentiobiose unit (between −0.9 and −1.1 a.u.) are higher than the partial charge of oxygen in the water molecules (−0.7 a.u.), and the charge of the H atom of the OH^•^ radical (+0.4 a.u.) is higher than that in the water molecule (+0.34) (Appendix A). However, note also that due to thermodynamical barriers and despite this electrostatic attraction, the OH^•^ radical cannot oxidize the sugar moieties and will rather migrate from site to site on their surface. This favors the encounter and oxidation of the neighbor –C_a_H_3_, rather than –C_b_H_3_ (Figure 13). The *sugar-driven* probabilities of R^•^C_a_ formation becomes: ^2^/_3_ + ^1^/_6_ = ^5^/_6_ and of R^•^C_b_: 0 + ^1^/_6_ = ^1^/_6_. The corresponding spectra are labeled “^5^/_6_ ^1^/_6_” in Figure 10 and Figure 12.

According to this discussion, the formation probabilities of the R^•^C_a_ and R^•^C_b_ products should lie between the extreme cases: (^1^/_2_ ^1^/_2_) (no interference of the sugar substituents) and (^5^/_6_, ^1^/_6_) (sugar-driven oxidation of –C_a_H_3_). Obviously, this probability ^5^/_6_ has been roughly estimated and simply says that the R^•^C_a_ product is predominant. Figure 10 and Figure 12 suggest that this is the case. Note also that the cation, with its spectrum on Appendix A, is clearly not observed.

### 5.3. Absorption Spectra of Covalent Dimers

The R^•^C_a_ and R^•^C_b_ species may undergo covalent dimerization, through singlet pairing of their two unpaired electrons. In Figure 5b the dimer displays an intense band, with a slight redshift and an absorbance enhancement, with respect to crocin, and also a weaker band at 320 nm, with a ~50% absorbance enhancement with respect to the weak band of crocin at 250 nm. However, the corresponding dimers are too large for the present method of molecular simulation. We could perform simplified simulations, nevertheless, with dimers made of *frozen* monomers, taken in the optimized structures of the dimers, shown in Appendix A. This yields the spectra of Figure 14, where the spectrum of *frozen* Me_2_–crocetin, namely of its structure at 0 K, is added for comparison. 

In Figure 14 we also extended to the dimers our two hypotheses about the R^•^C_a_ and R^•^C_b_ monomers: if the monomers have the probabilities (½ ½), then the dimers must have the probabilities (^1^/_2_)^2^ = 0.25 for C_a_C_a_, 2 × (^1^/_2_)^2^ = 0.5 for C_a_C_b_ and 0.25 for C_b_C_b_; if the monomers have the probabilities (^5^/_6_ ^1^/_6_), then the dimers must have the probabilities (^5^/_6_)^2^ = 0.69 for C_a_C_a_, 2 × (^5^/_6_) × (^1^/_6_) = 0.28 for C_a_C_b_ and (^1^/_6_)^2^ = 0.03 for C_b_C_b_. Figure 14a shows that dimerization modifies the intense band of Me_2_–crocetin, with a little redshift and an intensity enhancement, in agreement with the measurement of Figure 5b. Figure 14b shows that the observed band at 330 nm is reproduced by the calculations. The ratio between the intensity of this band and that of the band at 250 nm of Me_2_–crocetin amounts to ≈2.6 for the (½ ½) combination of the monomers, to ≈1 for their (^5^/_6_ ^1^/_6_) combination, to be compared to the 1.5 factor of Figure 5b. This suggests that the predominance of the R^•^C_a_ product, as concluded from the results at shorter times, is still observed in the spectra of their dimers, and that the ^5^/_6_ probability of this product is probably overestimated. Actually, the spectra of the dimers would deserve more rigorous simulations.

### 5.4. Absorption Spectra of Products of Radical’s Disproportionation

In γ-radiolysis (at a weak dose rate), the final products (Figure 6b) have a spectrum much different from that of radical dimers. One significant difference in γ radiolysis, compared to pulse radiolysis, is the low concentration of radicals and the formation of van der Waals complexes, (crocin, crocin (–H))^•^ between radicals and non-oxidized crocin (Reactions (7) and (8)). 

Investigation of such complexes is at the limit of our present methods because of the size of the systems involved. We first considered the complexes (Me_2_–crocetin, Me_2_–crocetin(–H))^•^. The optimized structures of the complexes of the R^•^C_a_ and R^•^C_b_ radicals are shown in Appendix A. The complexes display a *parallel* arrangement of the polyene chains, thus achieving mutual solvation of these hydrophobic moieties. The formation free energies of these complexes are very negative (−0.5 eV) (Table 2). This means that if such a complex forms, then it does not separate. Unfortunately, these structures are disappointing because the encounter of two such complexes can provoke either disproportionation, if the H transfer occurs, or dimerization as well, if one radical center, C_a_H_2_ or C_b_H_2_, of one of the complexes, comes close to the radical center of the other complex. However, this possibility of dimerization should be reduced by bulky substituents. With this in mind, we introduced glucose moieties and optimized a few structures of the (Glu_2_-crocetin, Glu_2_-crocetin(–H))^•^ complexes. At this stage, we do not claim to have a general view of such complexes. One typical structure involving the R^•^C_a_ radical is shown in Appendix A. Note that this structure is ruled by parallel polyene chains, like in the Me_2_–crocetin complex, and also by hydrogen bonds between neighbor glucose moieties. 

If now two such complexes are encountered, it seems clear that a parallel arrangement is hindered by the substituents, and that crossed structures, with an X shape or a T shape, will be rather favored. Then, the dimerization vs. disproportionation alternative occurs if, in the resulting dimer of complexes, both radicals are in contact inside the structure. If these radicals are both of the predominant R^•^C_a_ type, then the crossed shape will forbid dimerization, and allow H atom transfer and disproportionation. If one of these radicals is of the rare R^•^C_b_ type, dimerization cannot be excluded but will be rare as well.

It can be seen in Table 2 that disproportionation involves two steps: first transfer of one H atom from one radical to the other one, yielding a *primary,* twice oxidized species, then cyclization of this species. In the following, we note Me_2_–crocetin(–2H) C_x_C_y_ the product of disproportionation, where the C_x_ and C_y_ carbon atoms have been oxidized. It can be seen also that in most cases the primary species is a closed shell molecule with alternate single and double CC bonds between two =CH_2_ groups. This is the case for the Me_2_–crocetin(–2H) C_a_C_a′_, C_a_C_b′_, or C_b_C_b′_ species, see Appendix A. This is due to an odd number of bonds between the two oxidized carbon atoms (Figure 1). Table 2 shows that the formation free energies of these species amount to about −0.3 eV. In one case only, Me_2_–crocetin(–2H)C_a_^•^C_b_^•^, the primary species is not a closed shell molecule, because of an even number (6) of CC bonds. This species is a diradical, made of two –CH_2_^•^ groups, for which the triplet state can be calculated by DFT, but the singlet state cannot. In Table 2 we consider that in this case, the triplet state is more stable than the singlet, thus suggesting that the formation free energy of this singlet species is positive, and the species inaccessible. In any case both –CH_2_ groups can come close to each other and fuse as a –CH_2_–CH_2_– bond, leaving a cyclic, closed shell molecule (Appendix A). Note that cyclization is also possible for the *unexpected* Me_2_–crocetin(–2H) C_a_C_b_ system. It is clear that the *primary* twice oxidized radicals are transients and cannot be observed in γ radiolysis.

In Figure 15 the absorption spectra of the four possible cyclic products are shown: Me_2_–crocetin(–2H) C_a_C_a′_, C_a_C_b,_ C_a_C_b′_, and C_b_C_b′_, and of three of their possible combinations. The spectra of the three accessible transients, primary products are also shown in Appendix A. 

The mechanism of the crocin oxidation by γ-radiolysis may be derived from the best fitting between the experimental absorption spectrum of the product in Figure 6b and the simulated spectrum of Figure 15 and Appendix A. Figure 15a shows that, in contrast with Figure 6b, the cyclic C_a_C_b_ (or the symmetric C_a′_C_b′_) molecule has an intense absorption spectrum similar to that of Me_2_–crocetin (with a 15 nm blue shift). The three other *accessible* cyclic products, C_a_C_b′_, C_a_C_a′_, and C_b_C_b′_, have rather close spectra very different from that of Me_2_–crocetin and with only weak bands in the 400–450 nm zone. If we consider the ^1^/_3_ ^1^/_3_ ^1^/_6_ ^1^/_6_ combination, assuming that the R^•^C_a_ and R^•^C_b_ species have the same probability and that the C_a_C_b_ cyclic is also present, this spectrum would have still an intense band at 425 nm, which is clearly not observed. The 0 ^1^/_2_ ^1^/_4_ ^1^/_4_ combination makes the same assumption on the R^•^C_a_ and R^•^C_b_ species but discards the C_a_C_b_ cycle. This spectrum, with a weak band at 321 nm and a still weaker band at 407 nm is in fair agreement with the observed spectrum. The 0 ^1^/_2_ ^5^/_12_ ^1^/_12_ combination also discards the C_a_C_b_ cycle and reflects the probabilities ^5^/_6_ and ^1^/_6_ for R^•^C_a_ and R^•^C_b_. Actually, the corresponding spectrum, with a weak band at 323 nm and a still weaker band at 414 nm, is also very close to the observed spectrum because the spectra of the involved cyclic products are close to each other. Figure 6 shows that after γ radiolysis, the absorbance of the sample is drastically reduced, up to ≈^1^/_9_ of that of crocin at 441 nm. Figure 15b shows that the simulations well reproduce this fact, with the same factor of ^1^/_9_. Appendix A presents the spectra of the *transient* primary species and their 0 ^1^/_2_ ^1^/_4_ ^1^/_4_ combination. It can be seen that all the spectra are different from those of the final cyclic molecules and that their combined spectrum is clearly not observed. 

In brief, this section provides the interpretation of the observed spectrum after γ radiolysis (weak dose rate). The Me_2_–crocetin(–2H) C_a_C_b_ product is not observed. The three *accessible* cyclic products are observed through their superposition, with a weak band at ≈320 nm, to be compared with the observed one at ≈330 nm, and a weaker low energy band at ≈410 nm to be compared with the observed one at 430 nm. The drastic absorption coefficient collapse after γ-radiolysis is well reproduced, but the spectra cannot help discriminate the assumptions about the probabilities of the R^•^C_a_ and R^•^C_b_ species.

The critical point is that the R^•^C_a_ species is predominant and has a radical center close to a bulky substituent. This prevents the close approach of two reactive –C_a_H_2_ groups to each other, and thus from a covalent dimerization. Note that this is true only if (crocin, crocin (–H)^•^ radical) complexes predominantly exist, with high steric requirements. In pulse radiolysis, such complexes do not exist, the approach of the radicals is ruled by chance and the sugars may step aside and give way to dimerization.

## 6. Conclusions

To understand the detailed mechanism of the antioxidant properties of crocin, observations of transients to the end-products of the crocin oxidation were performed by the radiolysis method, in pulse or steady-state regime, and crossed with demanding molecular simulations.

The high value of the rate constant of the crocin oxidation by the OH^•^ radical confirms the efficient direct antioxidant properties of crocin, thus protecting the neighbor molecules from oxidation, and from the deleterious effects of possible induced chain reactions with oxygen. Moreover, the thermochemical calculation on the model Me_2_–crocetin indicates that by its redox potential, crocin may behave also as an indirect antioxidant in restoring, through an H-atom transfer, neighbor molecules from their oxidized radical. Note that the present radiolysis and simulation studies are restricted to the oxidative attack by one OH^•^ radical per molecule. Moreover, the stable products of the reaction still possess several similar oxidable sites that enhance the antioxidant power of this kind of molecule.

The simulated spectra, which reproduce fairly well the spectra obtained by radiolysis, gave essential insights into the molecular structures and the detailed mechanisms of oxidation. The observed optical absorption spectrum of crocin, mostly constituted of an intense asymmetrical band, is interpreted with the help of conformation analysis. This analysis emphasizes the predominant influence of the polyene chain conformers on the optical spectrum. Significantly, they may also explain the similar ones of numerous carotenoid spectra, mainly the same intense and asymmetrical band at around 440 nm.

The optical spectrum of the radical resultin*g* from the crocin oxidation by OH^•^ radical is constituted of a specific band at 678 nm in addition to the main asymmetrical band, similar to crocin at 441 nm but slightly less intense. The thermochemical calculations and the molecular simulations of the optical spectra provide an assignment of the site of the OH^•^ radical attack on crocin. It is important to note that this OH^•^ attack, contrary to other carotenoids, occurs only on the polyene chain. Interestingly, the sugar sites of crocin do not scavenge but just attract electrostatically OH^•^ radicals, and thus play an important role in favoring the H-abstraction from methyl groups of the chain, predominantly from the close neighbor ones (*sugar-driven* mechanism).

After a pulse (of high dose rate), the radicals dimerize into a stable covalent dimer. During the gamma irradiation (of lower dose rate), the radicals first are complexed by excess crocin molecules before their disproportionation into a twice-oxidized crocin and a crocin molecule. From molecular simulations, the oxidized molecules may display various cyclic conformations.

## Figures and Tables

**Figure 1 antioxidants-12-01202-f001:**
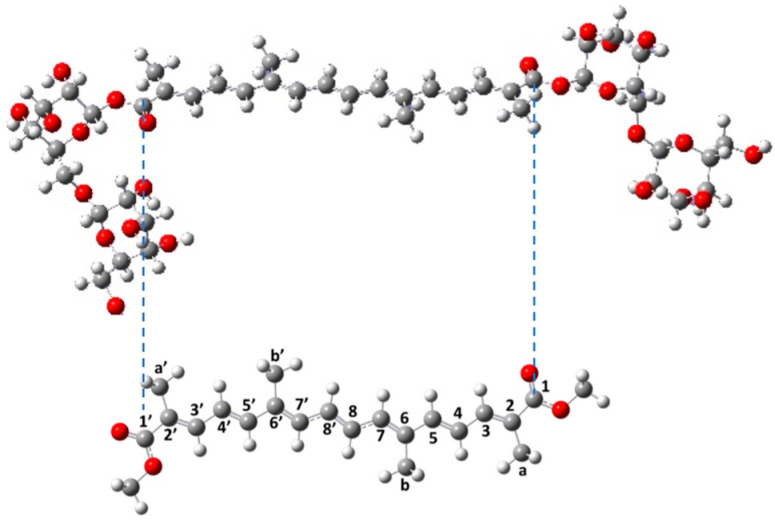
(**Top**) The *all-trans* crocin molecule. (**Bottom**) The *all-trans* Me_2_–crocetin molecule used as a model for the molecular simulations. The numbering of the carbon atoms is indicated on the polyene chain (between the dotted lines).

**Figure 2 antioxidants-12-01202-f002:**
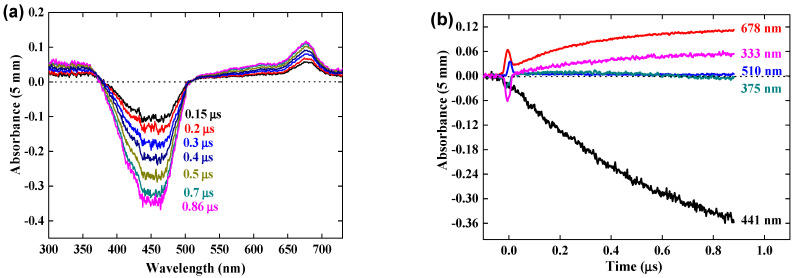
(**a**) Transient difference spectra within 1 μs after the pulse for ([croc] = 0.0385 mM). (**b**) Kinetics at 333, 375, 441, 510 et 678 nm. Dose = 97 Gy. Optical path: 0.5 cm.

**Figure 3 antioxidants-12-01202-f003:**
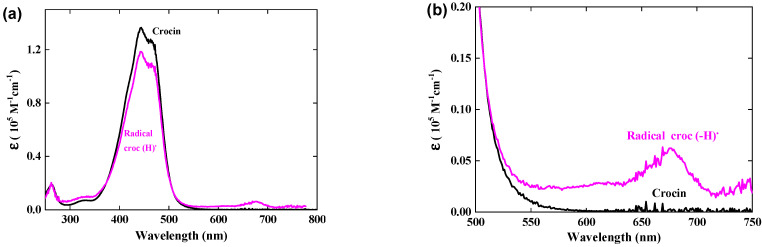
(**a**) Normalized optical absorption spectra in the molar absorption coefficient of the crocin (in black) and of the oxidized radical of crocin (in pink). (**b**) Zoom of both spectra between 500 and 750 nm, with an isosbestic point at 510 nm.

**Figure 4 antioxidants-12-01202-f004:**
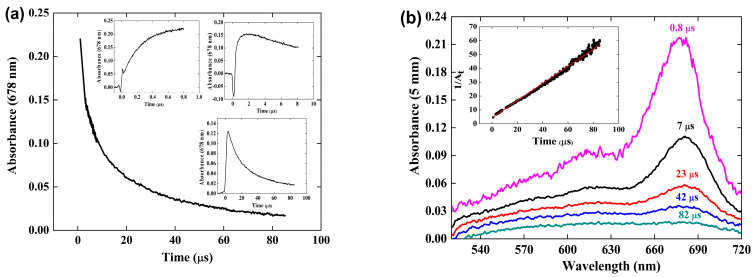
(**a**) Kinetics at 678 nm within 84 μs for [croc] = 0.13 mM, reconstituted according to the absorbance at 1 μs (Appendix A) and to the signals over 10 and 100 μs (insets). (**b**) Transient spectra within 84 μs for ([croc] = 0.13 mM. Inset: Second-order test.

**Figure 5 antioxidants-12-01202-f005:**
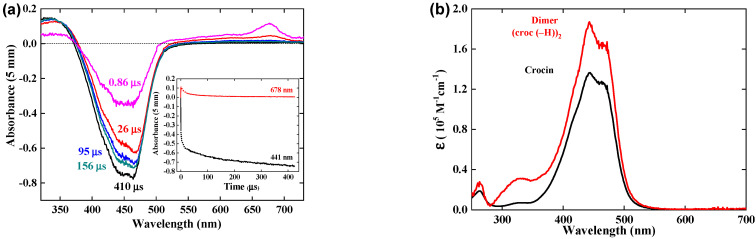
(**a**) Time-resolved spectra within 20 and 418 μs. The spectrum at 0.86 μs (in pink) of Figure 2a is added for comparison. Inset: Kinetics at 441 nm and 678 nm. Data before 0.86 μs are taken from Figure 2b. (**b**) Normalized optical absorption spectra in t molar absorption coefficient of the crocin (in black) and of the radical dimer (croc (–H))_2_ (in red).

**Figure 6 antioxidants-12-01202-f006:**
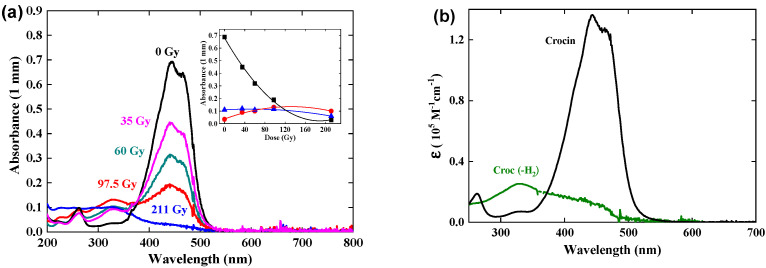
(**a**) Dose-dependent optical absorption spectra of a γ−irradiated crocin solution. Inset: Variation of the absorbances at 330, 375, and 441 nm versus the dose. [croc] = 0.057 mM. Dose rates = 3.5, 6.0, 9.8, or 21.1 Gy/min. (**b**) Spectra of the crocin (in black) and of its final oxidation product (in green) arising by γ-irradiation and obtained from the deconvolution, both calibrated in molar absorption coefficients.

**Figure 7 antioxidants-12-01202-f007:**
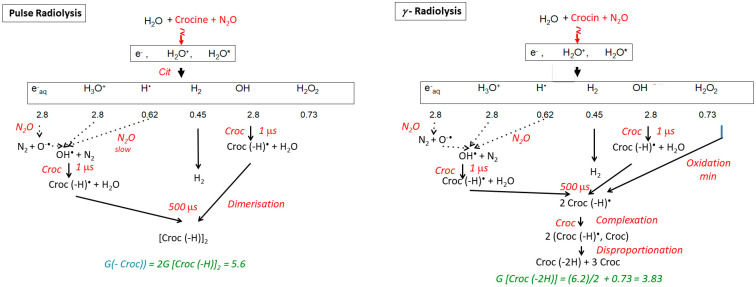
Schemes of the pulse and gamma radiolysis of aqueous solutions of crocin saturated by N_2_O. The yield values are in 10^−7^ mole J^−1^ units.

**Figure 8 antioxidants-12-01202-f008:**
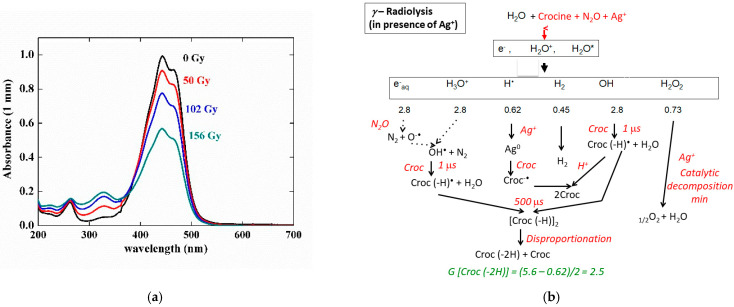
(**a**) Absorption spectra evolution with the γ-dose of a mixed solution of 0.084 mM of crocin and 0.1 M of silver perchlorate saturated by N_2_O. (**b**) Scheme of the radiolysis of the solutions. The yield values are in 10^−7^ mole J^−1^ units.

**Figure 9 antioxidants-12-01202-f009:**
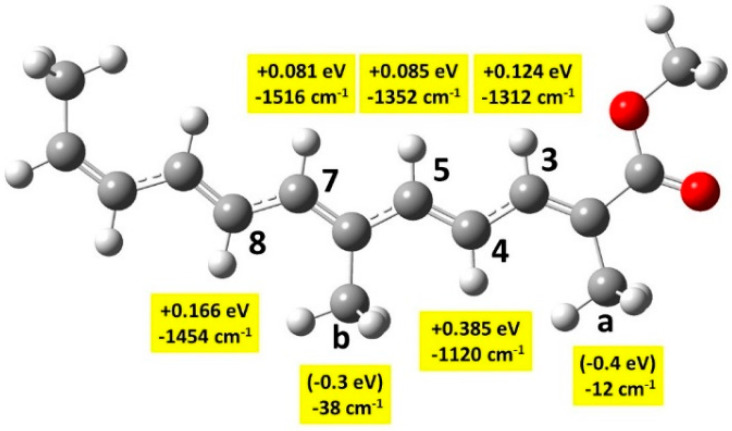
Model 1 for the evaluation of transition states for H atom abstraction by the OH^•^ radical at the B3LYP/apvdz/SMD level. Barrier heights (eV) and imaginary frequencies (cm^−1^), written with a negative sign. Bracketed numbers for C_a_ and C_b_ are approximate but the negative sign is ensured (see text).

**Figure 10 antioxidants-12-01202-f010:**
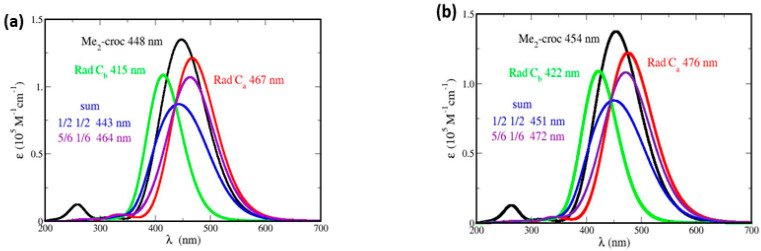
Absorption spectra of Me_2_ crocetin, of its R^•^C_a_ and R^•^C_b_ radicals and their sum with two coefficients, with fwhm = 0.50 eV, spin screening 20% and (**a**) the small pSDD and (**b**) the larger pSDD+ gaussian bases.

**Figure 11 antioxidants-12-01202-f011:**
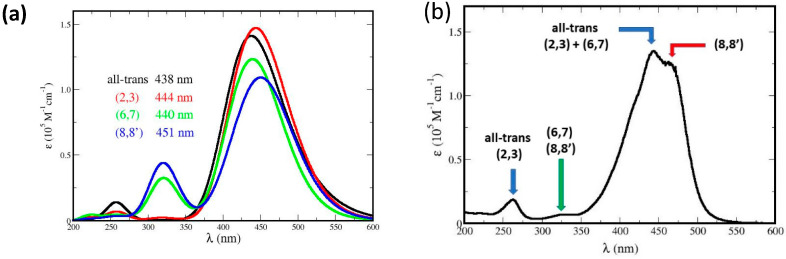
(**a**) Absorption spectrum of a few Me_2_–crocetin conformers, yielded by MC simulations with *all-trans* and bent constraints; with fwhm = 0.50 eV. (**b**) Measured absorption spectrum of crocin (Appendix A) with structure propositions according to the Me_2_–crocetin results of Figure 11a.

**Figure 12 antioxidants-12-01202-f012:**
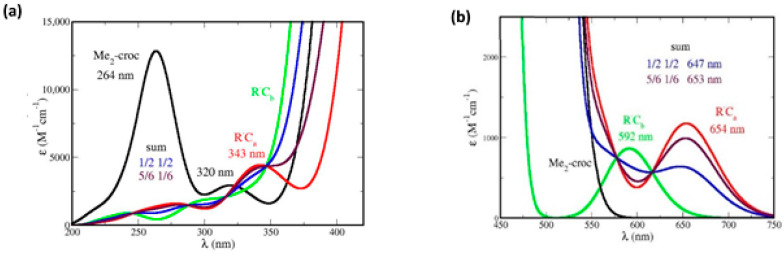
Simulated spectra of the R^•^C_a_ and R^•^C_b_ species with the large *pSDD+* basis. The focus is made on weak bands at (**a**) higher energies with *fwhm* = 0.50 eV, and (**b**) at lower energies with *fwhm* = 0.20 eV and s*pin screening* 20%.

**Figure 13 antioxidants-12-01202-f013:**
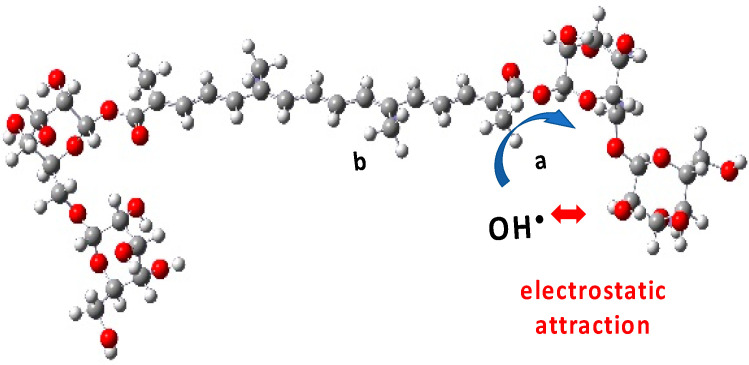
Despite the electrostatic attraction, the OH^•^ radical cannot oxidize the sugar moieties of crocin. This *sugar-driven* mechanism favors the oxidation of the neighbor –C_a_H_3_, rather than –C_b_H_3_.

**Figure 14 antioxidants-12-01202-f014:**
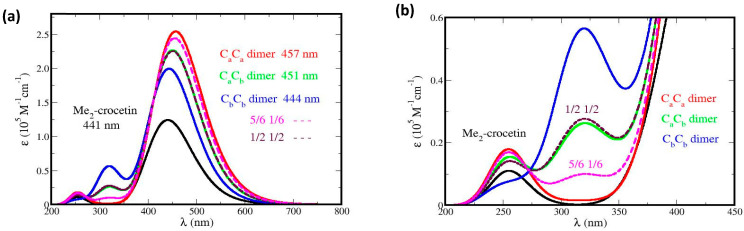
Absorption spectra of the covalent dimers through simulation of frozen monomers, with *fwhm* = 0.70 eV. The focus is made on (**a**) the intense bands, and (**b**) the weak, high-energy bands. The labels (½ ½) and (^5^/_6_ ^1^/_6_) refer to the monomer probabilities.

**Figure 15 antioxidants-12-01202-f015:**
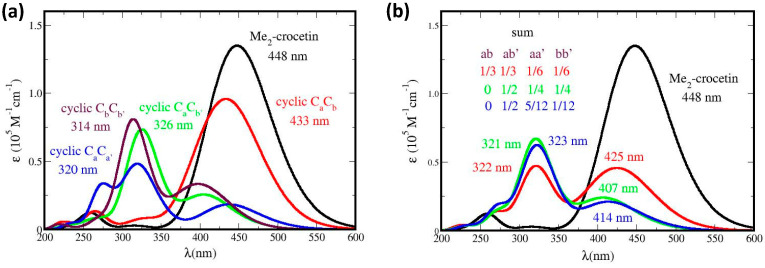
Absorption spectra of (**a**) the four possible cyclic products of the radical’s disproportionation, and (**b**) three of their combinations, with *fwhm* = 0.50 eV.

**Table 1 antioxidants-12-01202-t001:** Radiolytic yields (in 10^−7^ mole J^−1^) and molar absorption coefficients (in 10^5^ M^−1^ cm^−1^) of the crocin oxidation products ([croc] = 0.13 mM).

Solution	Crocin (Pulse Radiolysis)	Crocin (γ-Rays)	Crocin + Ag+ (γ-Rays)
**G** _exp_ **(- croc)**	-	4.0	2.1
**G** _calc_ **(- croc)**	6.2	3.84	2.5
**G_calc_(croc –(H^•^)**	6.2	-	-
**ε_678nm_(croc –(H)^•^)** **ε_441nm_(croc –(H)^•^)** **ε_333nm_(croc –(H)^•^)**	0.0623 1.17 0.10	-	-
**G_calc_(croc –(H))_2_**	3.1	-	-
**ε_333nm_(croc –(H))_2_** **ε_441nm_(croc –(H))_2_** **ε_678nm_(croc –(H))_2_**	0.31 1.84 0	-	-
**G_calc_(croc (–2H))**	-	3.84	2.5
**ε_333nm_(croc (–2H))** **ε_441nm_(croc (–2H))**	-	0. 26 0.13	-

**Table 2 antioxidants-12-01202-t002:** Molar reaction free energies (in eV) for processes occurring in the oxidation of Me_2_–crocetin by the OH^•^ radical. R^•^ = (Me_2_–crocetin (–H))^•^ radical, with one oxidized carbon atom. The carbon atoms C_a_ and C_b_ are shown in Figure 1.

Reaction			∆_r_G (eV)
**Me_2_–crocetin + OH** ** ^•^ **	**→ R^•^C_a_**	+ H_2_O	−1.98
	→ R^•^C_b_		−1.94
R^•^C_a_ + R^•^C_a_	→ dimer C_a_C_a_	-	−1.09
R^•^C_a_ + R^•^C_b_	→ dimer C_a_C_b_	-	−1.05
R^•^C_b_ + R^•^C_b_	→ dimer C_b_C_b_	-	−1.12
R^•^C_a_ + R^•^C_a_	→ Me_2_–crocetin (–2H) C_a_C_a′_ ^1^A	+ Me_2_–crocetin	−0.26
	→ Me_2_–crocetin (–2H) C_a_C_a′_ cyclic		−0.48
	→ Me_2_–crocetin (–2H) C_a_^•^C_b_^• 3^a		+0.30
	→ Me_2_–crocetin (–2H) C_a_^•^C_b_^• 1^A		≥+0.30
	→ Me_2_–crocetin (–2H) C_a_C_b_ cyclic		−1.28
	→ Me_2_–crocetin (–2H) C_a_C_b′_ ^1^A		−0.26
	→ Me_2_–crocetin (–2H) C_a_C_b′_ cyclic		−0.39
R^•^C_a_ + R^•^C_b_	→ Me_2_–crocetin (–2H) C_a_C_a′_ ^1^A	+ Me_2_–crocetin	−0.30
	→ Me_2_–crocetin (–2H) C_a_C_a′_ cyclic		−0.52
	→ Me_2_–crocetin (–2H) C_a_^•^C_b_^• 3^a		+0.26
	→ Me_2_–crocetin (–2H) C_a_^•^C_b_^• 1^A		≥+0.26
	→ Me_2_–crocetin (–2H) C_a_C_b_ cyclic		−1.32
	→ Me_2_–crocetin (–2H) C_a_C_b′_ ^1^A		−0.30
	→ Me_2_–crocetin (–2H) C_a_C_b′_ cyclic		−0.43
	→ Me_2_–crocetin (–2H) C_b_C_b′_ ^1^A		−0.29
	→ Me_2_–crocetin (–2H) C_b_C_b′_ cyclic		−1.04
R^•^C_b_ + R^•^C_b_	→ Me_2_–crocetin (–2H) C_a_^•^C_b_^• 3^a	+ Me_2_–crocetin	+0.22
	→ Me_2_–crocetin (–2H) C_a_^•^C_b_^• 1^A		≥+0.22
	→ Me_2_–crocetin (–2H) C_a_C_b_ cyclic		−1.36
	→ Me_2_–crocetin (–2H) C_a_C_b′_ ^1^A		−0.33
	→ Me_2_–crocetin (–2H) C_a_C_b′_ cyclic		−0.47
	→ Me_2_–crocetin (–2H) C_b_C_b′_ ^1^A		−0.33
	→ Me_2_–crocetin (–2H) C_b_C_b′_ cyclic		−1.08
R^•^C_a_ + Me_2_–crocetin	(R^•^C_a_, Me_2_–crocetin)	-	−0.52
R^•^C_b_ + Me_2_–crocetin	(R^•^C_b_, Me_2_–crocetin)	-	−0.52

## Data Availability

The data that support the findings of this study are available on request from the corresponding authors (M.M. and P.A.).

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
