# Peer review of "Unveiling the Intimate Mechanism of the Crocin Antioxidant Properties by Radiolytic Analysis and Molecular Simulations"

_antioxidants, 2023, doi:10.3390/antiox12061202_

Round 1
Reviewer 1 Report
The manuscript has been well desging, however, there are some minor revision as follows:
- The references should be updated, references 7-8-9-10-11-12-13-14 could be updated.
- I recommend to use the relevent articles, which they published recently in Antioxidants MDPI Journal like: https://doi.org/10.3390/antiox11112213
- The conclusion should be summarized.
- City and country of all the companies of all the reagents and equipment’s employed must be mention. In case of USA companies, include the city and the state abbreviation. Unify and apply to the entire document.
The quality of English Language is acceptable.
Author Response
Minor editing of English language required.
Some repetitions have been suppressed in the last paragraph of the Introduction. See the revised version.
The references should be updated, references 7-8-9-10-11-12-13-14 could be updated. I recommend to use the relevent articles, which they published recently in Antioxidants MDPI Journal like: https://doi.org/10.3390/antiox11112213
Line 38. New references 10 and 11 have been added. Lines 73-75. A new reference 28 has been added.
The conclusion should be summarized.
Among the great amount of results and discussions of the radiolysis and the molecular simulation study, we strictly selected in the Conclusion only the most original aspects.
City and country of all the companies of all the reagents and equipment’s employed must be mention. In case of USA companies, include the city and the state abbreviation. Unify and apply to the entire document.
Line 95-97. City and country of all the companies of all the reagents and equipment have been added. The ELYSE equipment was home-made constructed by our laboratory ICP as mentioned by references 29-30.
Reviewer 2 Report
Carotenoids are health promoters. Due to their long conjugated chains, carotenoids are highly reactive and efficient scavengers of free radicals. This research topic about crocin antioxidant properties is very interesting and desirable. It is generally well-constructed. However, some points must be clarified before the article can be accepted.
1. 1. What was the temperature of the crocin aqueous samples used for the described experiments? Was it room temperature?
2. 2. According to the literature data, the oxidation potential of a carotenoid in a nonpolar environment was higher than in a polar environment. Was the effect of the environment (solvent type) on the reaction mechanisms examined? Furthermore, dimethylsulfoxide is usually applied as a solvent for carotenoid antioxidant studies. Consequently, the comparison (towards other carotenoids, including polar carotenoids) is more accessible with the previously published data.
3. 3.. What was the analysis's precision and accuracy?
Author Response
What was the temperature of the crocin aqueous samples used for the described experiments? Was it room temperature?
Line 110. The temperature of experiments has been given.
According to the literature data, the oxidation potential of a carotenoid in a nonpolar environment was higher than in a polar environment. Was the effect of the environment (solvent type) on the reaction mechanisms examined? Furthermore, dimethylsulfoxide is usually applied as a solvent for carotenoid antioxidant studies. Consequently, the comparison (towards other carotenoids, including polar carotenoids) is more accessible with the previously published data.
Lines 85, 118 and 403. The approach of this time-resolved study is basically to use as oxidant the source of OH• radicals produced instantaneously and homogeneously during the radiolysis of water. The words ‘in water’ have been three times added.
Moreover, in the present study, we do not compare the redox potential of crocin, but that of its radical E0(croc (–H2)) / (croc (–H))• (which is never given in the literature) with the known value of the Ag+ potential, both in water. Effectively, the same order, with a positive shift of the values, would be expected in a non-polar environment.
What was the analysis's precision and accuracy?
Line 113. The uncertainty on the absorbance measurements has been given.